# Sustainable Financial Risk Modelling Fitting the SDGs: Some Reflections

Christian Walter [1,2] 

1   Fondation Maison des sciences de l'homme (FMSH), 75006 Paris, France; christian.walter@msh-paris.fr
2   Department of Economics and Finance, Kedge Business School & SDSN France, 33405 Talence, France

**Abstract:** This article argues that any ecological finance theory devised to fit the Sustainable Development Goals (SDGs) needs a paradigm shift in the morphology of randomness underlying financial risk modelling, by integrating the characteristics of "nature" and sustainability into the modelling carried out. It extends the common diagnosis of the 2008 financial crisis with considerations on the morphology of randomness and the reasons why neoclassical finance theory is not sustainable from this perspective. It argues that the main problem with unsustainable neoclassical finance risk modelling is its underlying morphology of randomness that creates a dangerous risk culture. It presents Leibniz's principle of continuity and Quetelet's theory of average as cornerstones of classical risk culture in finance, acting as a mental model for financial experts and practitioners. It links the notion of sustainability with the morphology of randomness and presents a possible alternative approach to financial risk modelling defined by rough randomness. If morphology of randomness in nature is properly described by fractal and multifractal methods, hence ecological finance theory has to include fractal properties into financial risk models. The conclusion proposes a new agenda for future research.

**Keywords:** fractal; ecology; sustainable finance; green finance; responsible regulation; scaling laws; power laws; principle of continuity; risk modelling; stochastic processes

## 1. Introduction

In the 2030 Agenda for Sustainable Development, in a part of the Sustainable Development Goals (SDGs), it is strongly asserted that "we are determined to protect the planet from degradation". In particular, "we are determined to take the bold and transformative steps which are urgently needed to shift the world onto a sustainable and resilient path". A quite striking example of an unsustainable and non-resilient path is given by finance and financialisation of the world economy [1] due to the financial approach of the neoclassical finance theory [2–4] because of the shortcomings of neoclassical finance theory in taking into account and meeting effectively the socio-economic and biophysical realities of the world. The financialisation of the economy due to neoclassical finance theory has caused considerable damage to the environment, with financial objectives taking precedence over sustainability objectives. This trend of environmental degradation was made possible because no environmental constraints were included in the measurement of financial risks to achieve social and environmental resilience. The issue of the relationship between financial risk and environmental risk was neglected as irrelevant in neoclassical financial theory. Thus, among the bold and transformative measures that are urgently needed to put the world on a sustainable and resilient path is the urgent need to rebuild finance on an ecological basis, to re-embed financial systems within ecological constraints and to develop an "ecological finance theory" [5]. An ecological finance theory will have to take over from neoclassical financial theory, by integrating the characteristics of "nature" and "sustainability" into the modelling carried out.

This paper is a piece for contributing to the abovementioned stake in order to interlock financial systems with the objectives of the 2030 Agenda. It is intended to be used as a platform for discussion between risk management practitioners in the financial industry and the regulator, as well as operators and scholars who have already applied investigative methods in other disciplines. Aligned with the [5] call, it is a proposition which presents a possible general framework for sustainable financial risk modelling. Although considerable work has been done in the area of financial risk modelling that takes into account the environment, sustainability and the greening of the financial system, much work remains to be done to develop integrated finance-nature models and "sustainable" or "green" metrics that can efficiently provide an overall picture for decision-makers. The issue is the following: As [5] points out, recent work on "green" finance generally superimposes a neoclassical financial approach on the examination of green issues, discussing them in terms of the investments and the money returns they provide, rather than questioning the capacity of financial systems to meet the challenge of sustainability [6]. According to this approach, "sustainability" or "green properties" are independent characteristics related to investor preferences, and only the impact of this feature on the changes in money metrics of "return" (usually modelled as a specific dependent variable) is analysed. Green financial instruments are conceived in this aim and the contrast in performance between "green" and "non-green" is discussed [7,8].

My position here is different. I view the system as a whole, and the ways in which we can make it sustainable from the metrics that compose it, as a "backbone". I propose to introduce a consistent ontological and epistemological framework that takes into account the interactions between the financial, socioeconomic and physical spheres; in particular, the impact of financial models in the making of reality [9–12] and the crucial role of financial risk modelling in this regard [13]. These works lead us to totally question the classical epistemology of the scientific approach and to construct an epistemology that takes into account the social and environmental impacts of mathematical models. This is the reason why I believe it is necessary to highlight the background philosophy that underlies neoclassical financial theory and the financial techniques that have resulted from it, in order to be able to better identify the non-financial philosophical principles at the root of the financialisation of the world. This will allow criticising this background philosophy in order to propose a change of philosophical paradigm more capable of combining finance and nature. This means that it is not simply a matter of using neoclassical finance theory, which will be "greened" by investor preferences and investment guidance, but of changing the very mathematical-technical structure of financial theory, i.e., changing the tools, the metrics. Inside the tools lies morphology of randomness. This paper considers that sustainability or unsustainability of metrics is related to the morphology of randomness that shapes the tools for risk calculations and modelling.

My approach is governed by the use of the philosophy and sociology of science with the goal of achieving improvement in the triple-bottom-line (economy, society and environment) by following a fully sustainable financial risk modelling approach. Consequently, the purpose here is not to do a research paper in financial mathematics or financial econometrics, but rather to propose a new research agenda by suggesting possible directions of research to re-embed nature into finance. Although not intended a priori for financial mathematicians or specialists in financial risk quantification but rather for managers or decision-makers and, more generally, those who are thinking about how to make finance closer to nature, this paper may nevertheless be of interest to them because it also encourages technical research in this area. It is meant to be a kind of "food for thought", a philosophical-driven proposition.

If neoclassical finance theory superimposes neoclassical financial logics on to the analysis of the environment, ecological finance theory calls upon us to turn this order upside down and to build new kinds of models. Among the new kinds of models to be developed, one of them is particularly important and sensitive—the financial risk modelling. I consider that a sustainable way to model financial risk should be a way to make the financial risk modelling "realistic", i.e., close to the actual characteristics of the risk to be modelled. This proposal is specified as follows: Some models are not "close" to the actual financial reality insofar as they do not take into account all the characteristics

of this reality (no taxes, equal interest for borrowers and lenders, no short constraints). This is not the point here, nor is it a question of exhausting the description of financial reality by ad hoc assumptions. We are interested in the similarity of the morphology of randomness of the models. We consider the probabilistic skeleton of the mathematical models and the philosophical background of this structure. The notion of "closeness to the real" designates a similarity between the morphology of randomness of the probabilistic model and the randomness structure of the real phenomenon to be modelled. I consider that a difference in randomness structure between a risk model and the reality of that risk is a cause of unsustainability. Reciprocally, a similarity between the morphology of randomness is a necessary but not sufficient condition of sustainability.

The challenge is therefore to understand the concrete characteristics of the risk to be modelled, in natural, economic or human environments. One of the main ideas of this article is that if financial risk modelling deviates too much from the real characteristics of environmental or human risks, then this modelling will not be sustainable, because it will not be realistic enough in this sense. Another idea of this article is that financial risk modelling takes place in an intellectual environment driven by financial or professional institutions, themselves infused with a behind-the-scenes philosophy about the morphology of randomness. Thus, the proposed methodology is a digression into the philosophy of science to better grasp the philosophical underpinning of financial risk modelling. The benefit of this methodology is that it allows a specific risk culture to be coupled with a particular philosophy of chance. A third idea of the article is that, in order to build a sustainable financial risk modelling, it is necessary to modify not only some mathematical techniques of financial risk modelling, but more generally the risk culture itself. To use an epistemological terminology borrowed from Kuhn [14], it could be said that there is a need for a paradigm shift in philosophy behind the financial risk modelling to build an ecological finance theory devised to fit the SDGs.

What are the possible avenues of research for new and sustainable financial risk modelling? Based on the principle of the proximity between financial risk modelling and the characteristics of environmental or human risk, one path among others appears promising, that which was opened up by Mandelbrot's fractal geometry. A fourth idea of this paper is to propose to further explore this avenue as possible, given the modelling successes that fractal geometry has encountered in the natural, life and social sciences. In fact, if one considers that sustainability needs to take account the "nature of nature" [15] and that to imitate the nature can "save the planet" [16–18], taking nature as model, nature as measure, and nature as mentor [19], one has to take account of a very specific "signature" of nature—its fractal geometry. To drive the argument to its salient edge, if the relevant geometry of nature is fractal, then financial risk modelling would have to be anchored on fractal representations in order to be in line with nature and become sustainable. This is not to say that fractal geometry would be the ultimate solution to all problems in sustainable financial risk modelling, but that this modelling approach seems promising enough to be considered seriously.

If one goes along with the abovementioned idea, assuming that nature is properly described by fractals implies that sustainable financial risk modelling should include fractal properties. The need of building risk models with fractal methods follows. A fifth idea of the paper is to show that these fractal properties already exist in neoclassical finance risk models, but only in a simple version. Thus, in order to move to a fractalisation of financial risk modelling, it is not necessary to "throw away" all the current research in finance, but only to extend certain aspects of existing models with more general fractals than the fractals of neoclassical finance. Consequently, one step which is urgently needed to shift the world onto a sustainable and resilient path could be the fractal generalisation of neoclassical financial risk modelling.

The outline of the article is as follows. Section 2 describes the influence of any financial risk modelling in the real economy, due to the performative power of risk models acting in technical tools and regulations, illustrating this with the example of Li's copula, the "Formula that Killed Wall Street". In an Anthropocene market society [20], it is extremely important to understand this phenomenon that a risk model provides a strong performative influence, as this performative influence

extends to the modelling of natural risks. This is the "financial ontology" of economic disasters. Section 3 enters this problem with a philosophical angle of the Leibnizian continuous principle combined with Quetelet's theory of averages as a pervasive mental model for neoclassical financial risk modelling, beyond the modelling techniques themselves, which since the 2000s have turned the tide of discontinuity. It introduces the connection between philosophy of science, mental model and risk culture in finance, and sketches a cartoon approach for sustainable risk culture in finance. It echoes Keynes' suggestion of two geometries in economics and introduces two risk geometries as two risk cultures in finance named "Brownian finance" and "non-Brownian finance", as two ways of thinking financial risk culture. Section 4 presents two important weaknesses in the risk culture of Brownian finance: the endogenous risk of crashes caused by the prudential regulation and the exclusion of large fluctuations as irrelevant and having to be modelled separately from "normal" fluctuations. Section 5 presents heuristically the fractal geometry and scaling laws as tools for devised sustainable structures adapted to nature and human geography (as the "law of 80/20"). The Conclusion proposes a new agenda for future researches.

## 2. Neoclassical Financial Risk Modelling: "We Were Wrong"

It is a well-documented fact today that one of the central problems in the 2008 financial crisis lay in a specific pricing equation—a mathematical formula that priced credit default swaps (CDS), the financial products supposed to provide financial protection against default risks—the mathematical copula devised by David Li and known as "Li's copula". This formula was faulty, not in the sense that the risks were miscalculated, but in the sense that they were mismodelled. Now, I will elaborate on this.

### 2.1. The Formula That Killed Wall Street

There have been many debates about this equation, which a famous article by Felix Salmon called a "Recipe for Disaster: The Formula That Killed Wall Street" [21–23]. The devil, it is said, is in the details. Playing with the words, I would say that, in this case, the detail was in the D-tails, D for "distribution" tails. The tails of a probability distribution describe the behaviour of a random variable in the zone that is far from its central value. Distribution tails can be "thin" (indicating a very low probability) or "fat" or "heavy" (indicating the opposite: a very high probability). Li thought that the risks of simultaneous credit default could be modelled using a mathematical coupling function called a "copula". However, he chose to use a probability of joint default based on Gaussian distribution (a Gaussian copula formula). The cognitive framework of the Gaussian distribution underestimates the probability of rare events, and creates the illusion that the risk is under control. The Gaussian copula formula encouraged excessive risk-taking because it gave precisely that cognitive illusion [24].

In [25,26], I analysed, in detail, a movie that is crucial to grasping the epistemological issues of financial risk modelling—Jeffrey Chandor's movie, *Margin Call* (2011). According to the philosopher Stanley Cavell [27], cinema can be, for us, a means of ethical teaching which does not use learned reasoning but only the power of images that touch us. The American philosopher Cora Diamond gave a reason for this: Images can often better convey ethical thinking than reasoned reflections. Images touch on an infra-rational level. That is what we are doing now, considering a particular scene from the movie *Margin Call*.

*Margin Call* tells the story of the fall of a merchant bank. The faulty risk modelling problem is very well presented with great epistemological accuracy in one central dialogue: a conversation between the risk management officer (Sarah Robertson) and the head of fixed income securities (Jared Cohen). This dialogue perfectly describes the epistemological and ethical issues carried by the probabilistic assumptions. The scene happens late at night, just after they have realised there is a problem with the mathematical risk assessment formula. They are beginning to assess the consequences of the wrong risk formula, in terms of potential losses for the bank. It is such an almost-perfect illustration of

the difference between risk calculation and risk modelling that it should be part of every course aiming to teach financial ethics. I will now quote the dialogue of Robertson and Cohen discussing Li's formula:

- − (Sarah Robertson) It's legit . . . the kid killed it. The formula's worthless.
- − (Jared Cohen) What does that mean?
- − (Sarah Robertson) It's broken.
- − (Jared Cohen) There are 8 trillion dollars of paper around the world relying on that equation!
- − (Sarah Robertson) Well, we were wrong.

This dialogue illustrates that a mathematical equation—here, Li's formula—led to financial disaster because it was ill-conceived ("the formula's worthless"). A simple, but ill-conceived, equation led to financial disaster.

In this sense, the financial crisis cannot be simplified in a "casino capitalism" approach, an expression which is a conflation of a particularly unhelpful kind [28], because it fails to help to understand the role of the risk modelling issue. Similarly, the financial crisis cannot be reduced to a massive misjudgement about the consequences of financial innovation [29] because, if there was a massive misjudgement, it concerns, first, the risk modelling issue. In neoclassical finance, something has not worked, which is not only linked to the avidity or greed of financial actors, but to the mathematical structure of financial risk modelling, the practices and techniques of a given risk culture carrying financialised tools. It is therefore of primary importance to investigate what, in the neoclassical modelling of financial risk, was inherently dangerous for the sustainability of the economy; for any ecological financial theory will have to start by getting rid of these dangerous elements and replace them with models adapted to sustainable aims.

In the next section, I adopt a Foucauldian approach (from the French philosopher and historian Michel Foucault) with the use of two important notions, that of "discourse" and "dispositive" [30]. I address the issue of the financialisation of economy through socio-technical devices ("dispositive") and risk modelling theories ("discourse") that legitimises them.

## 2.2. The Performativity of Financial Risk Modelling: Dispositive and Discourse

The idea that mathematical or financial models have an influence on reality leads to a very rich current of thought on the sociology of financial models, in particular the social studies of finance approach and the performativistic framework [9–12]. Ref. [31,32] are a good introduction to it.

### 2.2.1. Socio-Technical Instruments: The Financialised Tools

An important aspect of the performativity of mathematical financial risk modelling is the socio-technical dimension of the mathematical models. Financial instruments derived from mathematical models and financialised evaluation play an important role in the financialisation of the economy [33–35].

To better understand how a particular risk culture is created based on a probabilistic hypothesis, it is interesting to note a detail of the dialogue between Sarah Robertson and Jared Cohen in *Margin Call*: Robertson's remark "we were wrong". The use of the word "we" denotes a form of socially elaborated and shared knowledge with a practical aim, which helps to construct a culture of models common to a financial group, the culture of "how a model works" [36]. The culture of models is based on calculation and quantification conventions [37,38]. Quantification conventions ensure the same risk culture for financial practitioners. The culture of models is an "epistemic culture" in the sense of Knorr Cetina [39], specific to each group of financial practitioners: This culture diffuses a general way of thinking about technical objects. The technical objects of finance are overloaded with probabilistic techniques. Probabilistic techniques have had an enormous influence on risk assessment and the formation of quantification conventions [40]. Much work has been carried out on the basis of these methodological premises and I take the liberty of referring the reader to the references indicated so as not to lengthen the text excessively.

### 2.2.2. Discourse of Financialisation: The Financial Logos

I will now present the discourse of neoclassical risk modelling which I have called the "financial *Logos*" in [13]. I proposed to consider financial risk models as a "speech act" in Austin's sense [41,42]. For Austin, any statement must be considered as an action, which he names an "act of speaking", a "speech act". There are three dimensions to this act. A first dimension is that of the speech itself, the content of *what is said*. This is the dimension called "locutionary". The classic example is that of baptism. When the officiant pronounces the phrase "I baptise you", the words of this phrase represent the locutionary dimension of the speech. The second dimension expresses the desired effect of the speech, *which is done in saying so*. This effect is generally achieved if certain conventions are observed. In the case of baptism, the officiant must truly be a priest or pastor. Then, social conventions ensure that the baptism will have been performed. However, it can happen that unexpected effects, either unforeseen or unwanted by those who play the social convention of baptism, may occur, *what is done by the way of saying it*. For example, if someone in the assembly is an unbeliever and suddenly becomes unhappy that a new human being is baptised, his anger is an unintended and unwanted effect of the baptismal act of speech. Obviously, this is not what was desired by the participants in the baptism. Austin calls these unintended effects of the speech act (in the sense that they appear when there was no reason to believe that they would appear) the "perlocutionary" dimension of the act of speech.

To sum up, when we talk, three distinct things happen. There is what is said (the content of the discourse), what is done in saying so (the social conventions that ensure the performative effect of the discourse), and what is done by the way of saying it (the unexpected consequences of performativity). These three dimensions—locutionary, illocutionary, and perlocutionary—characterise a speech act.

The proposal I made to analyse the 2008 financial crisis is to consider the neoclassical financial risk modelling as a speech act. The three dimensions of this speech act (the *Logos* of financial risk modelling) are then as follows. The locutionary of the financial *Logos* is the mathematical content of financial risk modelling. Here, it is the mathematical writing of the probable that constitutes the "speech" on risk. The discourse on financial risk is taking shape and becoming performative because international financial regulations and globally accepted quantification conventions impose it on players in the financial world. Financial regulations and financial techniques represent the illocutionary dimension of the financial *Logos*. I argued that the 2008 crisis is an example of the unintended effect of the massive use of neoclassical risk modelling. In [26], I argued that this is the perlocutionary dimension of the financial *Logos*. My proposal was, therefore, to state that the discourse of the financial *Logos* embodied the unsustainable epistemological background of the paradigm of neoclassical finance theory.

My proposal makes it possible to combine the mathematical aspects of financial risk modelling with the social and conventional aspects of this writing of the probable. It allows us to understand how a reality imagined in mathematical finance research laboratories takes shape in the real economy. It allows us to understand how probabilistic hypotheses have an effect on decision-making processes and risk assessment. It allows us to include unintended effects, in this case, the consequence of the poor risk assessment of CDS, in the probabilistic speech act. It is in this sense that the probabilistic assumptions of neoclassical finance have made it possible to build a reality common to all financial practitioners.

### 2.3. Sustainable Finance and Epistemic Ethics

This way of thinking about the socio-technical influence of mathematical models raises ethical issues on epistemic responsibility in finance. This is because, as MacKenzie said at the end of *An Engine not a Camera*, "the notion of performativity prompts the most important question of all: What sort of a world do we want to see performed?" [31] (p. 275). This sentence echoes "what should be" from the ecological finance theory [5] and raises the question of epistemic ethics, i.e., taking into account the impact of the choice of model on the economy [43,44]. More precisely, the framing of decision-making by probabilistic assumptions exhibits a link between sustainable finance and financial

risk modelling issues. It is in this perspective that it is interesting to read again the work of the French physicist Pierre Duhem from the early 20th century.

In *The Aim and Structure of Physical Theory* (1906) [45], Duhem questioned the role of models in science. He puts forwards the following statement on the relationship between models and accidents: "We shall remind industrialists, who have no care for the accuracy of a formula provided it is convenient, that the simple but false equation sooner or later becomes, by an unexpected act of revenge of logic, the undertaking which fails, the dike which bursts, the bridge which crashes; *it is financial ruin when it is not the sinister reaper of human lives*" [45] (p. 93, emphasis added). One can see how Duhem's reflection is relevant to answer the ethical question raised by MacKenzie [31] and to address the issue of action raised in [5].

I am applying Duhem's reflection to financial risk modelling. Choosing a financial risk model means making an ethical choice. Li's formula was convenient, but not relevant as regards to the accuracy of the probabilistic assumptions in the case of risk modelling of neoclassical finance. The probabilistic assumptions of neoclassical financial risk modelling created the *hubris* and the decoupling between finance and economy, and gave rise to the 2008 crisis [24]. The short-termism of finance is strongly linked to the probabilistic assumptions embedded in the technical objects of everyday financial practice. As I was saying in the Introduction, the challenge then is to look for the underlying philosophy behind the probabilistic assumptions used in mathematical models of financial risk, which is the background philosophy on which they are built, and which constitutes a mental model for the financial world. I will now turn to this investigation in the following.

## 3. From the Philosophy of Science to the Risk Culture in Finance

A very important aspect of the probabilistic assumptions of neoclassical finance theory was the continuity of the stochastic processes used in the construction of financial models. Neoclassical view of financial risk was based on continuous stochastic processes. One consequence of the use of continuous stochastic processes is the importance given to the average in neoclassical finance. Continuity and average represent two intellectual foundations of neoclassical financial theory.

### 3.1. Mental Models in Finance

That continuity is a disabling property for financial modelling and that processes with discontinuities have to be implemented is something that is well known today in current research in finance, especially in financial risk modelling. Many mathematical models of financial risk are built on the basis of discontinuities ([46–53] among others, see [54] for taxonomy). However, this innovation is relatively recent, dating at most from the end of the 20th century and early 2000s. What I would like to clarify here is that, beyond the technical choices of ingredients for financial risk modelling (continuity or discontinuity), a philosophical background has existed during the 20th century in research in finance. This philosophical backgrounds acted as a "mental model" [55–57].

Mantzavinos [57] considers that, whatever their activity or intellectual aspirations, people must first ensure their material existence and, in order to do so, they must have mental models to act on. For Mantzavinos, one way to get out of the intellectual impasses of primitive mental models is to look at the side of science. It seems that all scientific disciplines come to the conclusion that there are no normative "facts" independent of the minds of the people who think them and whose interaction generates social norms. No phenomenon is beyond mental activity, whether individual or shared. Therefore, the question arises as to how to think about problem solving activity with mental activity. One of the answers is to consider that one can have a first approximation of mental representations by mathematical models, which constitute mental tools. These mental mathematical models shape the structure of the agents' expectations in front of the environment. A mental model can induce an information filter causing selective perception, a marker of which has been cognitive bias on statistical tests. A canonical example of a mental model producing a cognitive bias in the analysis of statistical tests in finance is given by the article by Granger and Orr [58] in which the authors truncate

the extreme values of the distribution to make the statistical results fit the Gaussian distribution [59]. Following this notion of a mental model, an underlying philosophy has been conveyed by the technical tools of practical finance.

Shared mental models lead to the emergence of norms. An epistemic framework on risk measurement has resulted from a shared mental model on uncertainty. For this cognitive reason, in [60] it is argued that one of the key puzzles of the preservation of the assumption of continuity in financial risk modelling could be illuminated by reference to familiar debates in philosophy of science. Although these philosophical debates may seem to be a scholastic preoccupation within a tight circle of specialists in philosophy of science, far from the financial stakes of risk modelling and with no impact on concrete financial practices, I argued, on the contrary, that the divergent positions about the mindset behind the financial risk modelling implicate entirely different views of what it was important to capture and how to model it. To put it another way, the scientific controversies over financial risk modelling—with or without continuity—could be illuminated by the choice of a background philosophy, which is extraneous to finance and which acts as a "philosophy hook". By "philosophy hook", I mean that a philosophical mental model prevented financial experts and academics from using more accurate mathematical models, relieved from the continuity or Gaussian assumption, as exemplified in [58]. This denial of financial risk models that are nevertheless more successful in terms of practical financial results resists the usual epistemological analysis and represents an "epistemological puzzle" in this respect [60]. The introduction of the notion of mental model helps to explain this epistemological puzzle.

*3.2. The Philosophy Hook: The Principle of Continuity and the Theory of Average*

The principle of continuity is a principle from natural philosophy postulating that in nature, things change gradually rather than suddenly. The most compact expression is found in Leibniz's words *Natura non facit saltus* (meaning "nature does not make a leap"). This principle was the source of differential and integral calculus as performed by Leibniz, then Newton. Note the ambiguous status of this principle, which can be understood either as mathematical or metaphysical. It provided the foundations for the ideas of Carl von Linné on classification of species, and later, Charles Darwin on the theory of evolution. It was then taken up by Alfred Marshall in his *Principles of Economics* (1890): "If the book has any special character of its own, that may perhaps be said to lie in the prominence which it gives to this and other applications of the Principle of Continuity." [61].

The principle of continuity—change is continuous—permeated all neoclassical economic models, which was the source of neoclassical finance theory. It was at the heart of the probabilistic assumptions in financial risk modelling and in this respect the financial risk modelling is an application of this continuity principle. The principle of continuity was the cognitive representation that governed academics' intuition in the mathematical financial risk modelling, in their work and in their teaching of finance. The same reasoning can be held for the principle of average. Quetelet is credited with developing a theory of the average man in the 19th century. This theory had an extraordinary influence on statistics in the 19th century, and then in the 20th century [62]. The index numbers, from which market indices are derived, are a consequence of this theory [63].

In the 20th century, both physics and genetics abrogated the principle of continuity. Quantum mechanics postulated discrete energy levels, and genetics took discontinuities into account. Yet economics, from which neoclassical financial theory originated, remained outside of these important intellectual transformations. Despite much evidence of economic phenomena that could not be explained by continuity, this principle remained in force until the end of the 20th century. Everything was going on as if, for mysterious reasons that had nothing to do with science, the principle of continuity was considered "natural" for finance and, in any case, preferable to discontinuous approaches.

It is accurate to say, at this stage of the overview, that as early as 1927, the principle of continuity was attacked by the Cambridge Italian economist Piero Sraffa [64]. Moreover, as early as the 1960s, the principle of continuity began to be challenged. In 1966, Wiener pointed out that "here some recent

work of Mandelbrot is much to the point. He has shown the intimate way in which the commodity market is both theoretically and practically subject to random fluctuations arriving from the very contemplation of its own irregularities is something much wilder and much deeper than has been supposed, and that the *usual continuous approximations to the dynamics of the market must be applied with much more caution* than has usually been the case, or not at all" [65] (p. 92, emphasis added). Wiener's criticism shows that the issue of continuity and the relevance of the average in the presence of strong irregularities were raised at an early stage. Very early on, comments challenged Marshall's attempt to impose continuity as a basis for economic modelling. What is interesting about Wiener's comment is that he does so from knowledge of Mandelbrot's work. Indeed, Mandelbrot was the first to assert that continuity and average were dangerous for financial risk modelling, and all his work was a development of how to take discontinuities into account.

At this stage of the presentation, an objection could be made. Could we not think of human markets and institutions as fundamentally different from those of nature and characterised by discontinuities that are completely unrelated to nature? In other words, how can we explain the relationship between a certain "continuous" conception of natural processes and the philosophical presupposition that we are trying to identify, which concerns financial markets? The answer to this objection was given by the statistical evidence of the same statistical properties of natural and social phenomena (see below). Thus, it seems that "natural" discontinuities and economic and social discontinuities may fall within the same mathematical framework of analysis. This resemblance raises the issue of an isomorphism between environmental risk and financial risk, a special case of "writing the book of the world" [66].

Before turning to Mandelbrot's conception of fractal-based financial risk modelling, let us look at how and where the principle of continuity and the theory of average influenced financial techniques.

### 3.3. From Philosophy to Mental Model of Financial Techniques: The Risk Culture of Finance

The principle of continuity contained methods of reasoning for financial practitioners derived from risk models based on the continuity assumption and, in this sense, the principle of continuity was at the core of a large number of financial techniques. The formulae developed by Fisher Black, Myron Scholes and Robert Merton in 1973, and later, the fundamental theorem of asset pricing of Harrison, Kreps and Pliska in 1979 and 1981, assume continuous price change. As MacKenzie and Spears [22] (p. 401) state: "It is the strategy of Black-Scholes modelling writ large: Find a perfect hedge, a *continuously-adjusted portfolio* of more basic securities that will have the same payoff as the derivative, whatever happens to the price of the underlying asset" (emphasis added).

The academic trajectory of the theory of average in finance was no less than that of the principle of continuity. This theory was the basis of all portfolio management techniques. The formulae for obtaining the mean-variance optimal portfolio of Harry Markowitz in 1959, and later, the Capital Asset Pricing Model of William Sharpe in 1964 are based on the relevance of the average. As with the principle of continuity, the theory of average was the cognitive representation that governed academics' intuition in the mathematical portfolio management, in their work and in their teaching of finance. As [67] shows, index-linked asset management is a clear and conceptually straightforward consequence of the use of the theory of average in portfolio management.

It has been said that neoclassical economic theory had constructed a mathematical utopia of the market [68–70]. I argue that this mathematical utopia was due to underlying philosophical preferences. Hence, to shed a light on Lawson's statement, "Contemporary academic economics is not in a healthy state" [69] (p. 3) and to answer the question asked by Chen, "What's wrong with economic math?" [68] (p. 17), I would say, the presupposition of the principle of continuity is coupled with the theory of average, with the whole forming the Brownian representation of finance and financial risk.

The abovementioned allows me to introduce a new idea: that of the cultural existence of two kinds of finance linked to two mental models, which I call "Brownian finance" and "non-Brownian" finance. By "Brownian finance", I mean a mental model to risk modelling in which the dynamics of price variations uses a motion (stochastic process) devised in 1827 by the Scottish botanist, Robert

Brown, named for this reason as "Brownian motion". Although in an implicit form and not quoted as such, the Brownian motion was introduced into finance by Louis Bachelier in his 1900 thesis on *Theory of Speculation* [71]. Brownian motion then appeared in physics with Einstein in 1905 and was then mathematically described by Norbert Wiener in 1923. The Brownian motion became the basic component of all equations of financial dynamics starting in the 1950s.

With the Brownian motion, financial engineering was able to develop sophisticated calculation techniques to evaluate the price of derivative products and resolve the question of contingent claims ("options") raised by Condorcet in 1782 [25] (p. 233). The solution to Condorcet's problem was the nobelized work of Black, Scholes and Merton in 1973. The Fourier heat equation was used to find the price of an option. Brownian finance is fundamentally smooth finance. There is no roughness in the paths, no fractures in markets and no financial crises. Brownian finance paints a picture of a regular, reassuring risk. With this mental model, financial risk definitely exists, but is not too unpredictable because it is smooth. Hence, the words "Brownian finance" describe a "mental model", that of a financial world in which financial risk is mentally regarded as something continuous which can be offset by diversification, thanks to the law of large numbers, and in which a "trend" (average risk) can be statistically identified.

By "non-Brownian finance", I mean the opposite. Non-Brownian finance describes a mental model in which the roughness dominates, in which stock market paths are made of jumps at all times and scales. Non-Brownian financial mental model incorporates jumps creating the "depth" of space, i.e., the depth of the risk. In non-Brownian finance, risk is perceived as "rough", like the curved space of non-Euclidian geometry. In non-Brownian finance, risk has "depth" and "relief", preventing the belief that it can be controlled simply by calculating statistical indicators, such as volatility. When the jump occurs, linear forecasts are found wanting. The stochastic processes of non-Brownian finance differ from Brownian motion as non-Euclidian geometry differs from Euclidian geometry. Non-Brownian stochastic processes paint a very different picture of risk, with much more surprises, as the path of the asset "jumps" all the time. The first stochastic processes that gave rise to non-Brownian finance were conceptualised by French mathematician Paul Lévy in 1924, and are named for this reason, "Lévy processes". Even if the time length between two points on a Lévy path decreases towards zero, the discontinuities remain. Lévy processes have discontinuous paths. A more intuitive way to express this is to say that the paths "jump" all the time, whatever the scale of analysis. Roughness is the essence of non-Brownian paths. Benoît Mandelbrot introduced Lévy processes in finance in the 1960s. In a nutshell, in the 20th century, the two scientists who gave birth to the two views of finance, the two risk cultures in finance, were Bachelier and Mandelbrot.

To perceive visually and in a direct intuitive way the difference between the two mental models, i.e., Brownian motion and Lévy processes, just consider the charts of the paths. Although both show irregular moves up and down, there is a clear visual distinction between the paths followed by the two processes. Although both trajectories have irregularities in upward and downward motion, and unpredictable up and down, the big difference between Brownian motion and Lévy processes lies in the visual aspect of the trajectory. It appears smooth in a Brownian motion, and rough in a Lévy process. Moves look "smoother" in Brownian motion than in a Lévy process, which have "jumps" or "discontinuities" almost everywhere at every scale. Following Mandelbrot's vocabulary, the difference between these two mental models is the difference between the "smooth" and the "rough".

I now introduce the notion of "risk culture" [72,73]. Risk culture is defined in different ways [74–76]. I use the definition given in IRM [77] (p. 6): "An effective risk culture is one that enables and rewards individuals and groups for taking the right risks in an informed manner" and in *Risk Culture's Critical Role in ERM*: "an organisation's 'risk culture' is the way in which its management and personnel collectively perceive and respond to risk". The risk culture had important consequences in the management of risks and the resulting unexpected losses. For example, as quoted in IRM [77] (p. 6), "In May 2012 JPMorgan Chase disclosed a multi-billion-dollar trading loss on its synthetic trading portfolio". By its own admission, the events that led to the company's losses *included inadequate understanding by*

*the traders of the risks they were taking*; ineffective challenge of the traders' judgment by risk control functions; weak risk governance and inadequate scrutiny" [78]. The risk culture was flawed.

I argue that the Brownian mental model of financial risk (continuity and average) shapes a particular hazardous risk culture. The reason is that, with a mental model built on continuity and average, financial risk disappears, since if prices change gradually and steadily, their future path is predictable and hedging can be found with financial derivatives. The principle of continuity coupled with the theory of average can be seen as a mental model which human action requires for the solution of practical problems [57]. This mental model is a model of a riskless economy due to the specific morphology of randomness. A riskless economy leads to a risk-free economy, an economy without limits. Moreover, an economy without limits, an economy in which risk is thought to have disappeared, is likely to become unsustainable. Thus the principle of continuity can be thought of as a philosophy hook that causes the risk culture to be unsustainable. The risk culture associated with the continuity principle is not sustainable because of its morphology of randomness.

The shift from this risk culture to another risk culture is an important issue for the implementation of any sustainable finance and ecological finance theory. This is because financial risk modelling designed to be sustainable must be realistic, i.e., it must take into account the actual risk characteristics of the economy and the environment. Let us elaborate on this point.

*3.4. Sustainable Risk Culture for Financial Risk Modelling*

Let us use a metaphor. In the environment, CO2 emissions due to human activity are considered toxic. CO2 is not a "green" gas. Similarly, I would say that Brownian motion is a toxic stochastic process for financial risk modelling. Just as CO2 impedes the sustainability of an industry, the risk culture of Brownian motion impedes the sustainability of the financial industry because the illusion of continuity can give birth to financial *hubris*: an excessive confidence in smooth risk modelling. The risk culture resulting from the use of Brownian motion does not allow us to imagine financial accidents. The morphology of randomness embedded in the Brownian risk culture does not allow us to consider the long run. Brownian finance is unsustainable for this reason.

Using the language resources of the vocabulary, I say that that the mental model of Brownian finance creates a "Brown finance" i.e., a "brown finance" as opposed to "green finance". In the same way that the goals of sustainability in industry are to eliminate CO2 emissions, one could say that the goals for sustainable financial risk modelling would be to eliminate the Brownian motion in all financial models. Then it will be a matter of building non-Brownian finance. To say it differently, the global conceptual framework of finance needs to be thoroughly reviewed in order to "detoxify finance" from the Brownian representation of financial risk modelling. While financial sector organisations have been under considerable pressure to change their risk culture [73,79,80], any risk culture that aims to be sustainable must be emptied of the Brownian representation of risk. This is also an answer to the 2008 call of IIF Final Report on Market Best Practices for Financial Institutions and Financial Products: "Cultivation of a consistent 'risk culture' throughout firms is the most important element in risk management" [81].

However, let us make it clear that when I state that finance must be emptied of the Brownian representation of risk, I am not saying that the entire field of finance has to be discarded, which would be much too excessive. What I mean by "finance rebuilt" is that epistemological assumptions about the morphology of randomness are unsafe for financial risk modelling. To be clearer, I argue that mathematical models of financial risk are based on an "underlying epistemological layer" related to the morphology of randomness which produces a specific risk culture, and that this epistemological hidden foundation needs to be modified. The "financial theory" is not "guilty" in itself, but the epistemological basis on which it is based. It is acknowledged that the current theory can and will be improved over time, but I argue that this improvement should begin with the replacement of the epistemological basis related to chance. I argue that the culture of risk stemming from the use of non-Brownian finance encourages caution at all times, and leads us to consider the financial fragility

of the systems to ensure their long-term survival. One of the reasons is that, when considering the Lévy measure of a given Lévy process, the part that remains "long run" is not the volatility (which vanishes through diversification) but the jump component, which determines the global pace of the movement [82].

A possible objection to this recommendation of changing the risk culture is that financial models are built on numerous assumptions, which we all know are not realistic. For example, it is often assumed no taxes, similar interest rate for lenders and borrowers, no constraints on short sales, etc. These assumptions do not hold in reality. However, even though we know that this may make them inaccurate, these assumptions enable us to construct solvable models, which provide valuable insights. The answer to this objection is the following. As we have seen above, an important aspect—documented but still relatively little taken into account—of mathematical models in the financial field is their performativity, i.e., their power to shape the "real". In finance, we are not in a situation where (according to this objection) "assumptions enable us to build solvable models" because solvable models are not an approximation of reality but the generator of that reality [31]. The question could therefore be: With which assumptions, with which risk culture, do we want to shape the world, to "provoke" reality? My point in this article is that the flaws of a risk culture based on Brownian finance are more serious than the flaws of a risk culture based on non-Brownian finance because continuity-based modelling has been at the root of financial *hubris* and recurrent financial failures since 1987. For Anthropocene societies and markets [20], what would be the "good" epistemic practices and instruments? What would be the "good" risk culture? I argue that a bad financial risk culture causes "real" disasters, due to the performativity of hazardous mathematical modelling because of hazardous morphology of randomness. This performative power of risk modelling can be understood as the "financial ontology" of disasters. To illuminate this idea with a practical example, let us look at below, the two pitfalls of unsustainable risk culture of Brownian finance.

## 4. The Unsustainable Risk Culture of Brownian Finance and Its Cognitive Biases

A good example of the need for changing the risk culture is made of the prudential regulation rules. It is known that multilateral institutions have acknowledged the need for a profound reform of the global financial system with emergence of Principles for Responsible Banking (PRB), Principles for Sustainable Insurance (PSI) and Principles for Responsible Investment (PRI). Although not mentioned in the previous list, responsible regulation is at least as important as responsible banking or responsible investment, and sustainable regulation is at least as important as sustainable insurance. However, the prudential regulations established after the 2008 crisis have had unexpected effects, just as dangerous as the ones they sought to address. The risk culture of Brownian finance is indeed not adapted to protect against financial accidents, and may even, through the regulation it carries, cause them. Black swans and extreme values can thus be understood as effects of the regulation of Brownian finance.

Furthermore, the current research in finance, specifically dealing with financial risk, is well aware of discontinuity and that there are many articles which try to overcome this issue. However, the status of discontinuities in Brownian risk culture is thought of as an "extra-feature" of the stochastic process, not as the core characteristic of the process itself [82].

### 4.1. Brownian Regulation and Financial Black Swans

Official reports have shown that international regulations have pro-cyclical effects and create potentially more dangerous financial situations than those that prevailed before the regulations were put in place. In recent work, it has been shown that neoclassical financial regulation has the potential to create the crashes it seeks to avoid and that there is, therefore, a "regulation risk", a risk created by the regulation itself [83]. This is because neoclassical regulation uses the Brownian risk culture to assess the capital requirement. This is not trivial and the reasons for this are explained below.

The objectives of international prudential regulation are to protect financial institutions from failures that could cause a damaging systemic risk to the economy. To this end, international prudential

regulation has established rules for calculating the minimum capital that financial institutions must have available to face market shocks. These rules have been formulated by the Basel Committee for the banks and by the European regulatory authorities for the insurance companies: respectively, Basel III and Solvency II. The problem was that the rules for calculating the capital requirement, e.g., Solvency Capital Requirement (SCR), had to use probabilistic assumptions to extrapolate the 10-day SCR from the 1-day SCR. The prudential rule says that the capital requirement for a 10-day horizon has to be calculated as the capital requirement for a 1-day horizon multiplied by the square root of 10. This is the "square-root-of-time-rule" for calculating minimum capital. This rule comes from the Brownian representation of financial risk modelling. The reasons for this extrapolation rule come from the properties of Brownian motion and will be explained in Appendix B. To say it differently, Brownian risk culture permeates international prudential regulation. The connection between mathematical assumptions (the square-root-of-time-rule) and the regulatory framework can be thought as the effects of the quantification convention that structures the world of neoclassical finance.

One of the consequences of the use of Brownian finance is the inability to deal with the problem of extreme values and the resulting intellectual cleavage in the appreciation of complex situations. A very good example of this inability to properly understand what is at stake is given by Alan Greenspan's editorial (16/03/08) in the *Financial Times* about the 2008 financial crisis: "We will never be able to anticipate all discontinuities in financial markets." Greenspan cannot imagine discontinuities being incorporated into a probabilistic description of the market. For Greenspan, (financial) nature does not make a leap and discontinuities are just unthinkable. Nevertheless, how to consider the discontinuities? The answer is given by Nassim Taleb [84]: Discontinuities are unpredictable "black swans". But [83] has shown that financial regulation of Brownian finance could create real market crashes. This is because the risk culture of Brownian finance prevents us from understanding the influence of continuity on the synchronisation of financial risk management practices.

We said earlier that the neoclassical representation of financial risk could be seen as the locutionary act of the financial *Logos* of neoclassical finance. Regulations that ensure the use of neoclassical probabilistic assumptions for risk measurement can be considered as the illocutionary act of financial *Logos*. Similarly, the production of unintended effects of regulation is akin to the perlocutionary act of financial *Logos*. Thus, my proposal is to interpret financial black swans as unexpected effects of regulation in the risk culture of Brownian finance [26].

### 4.2. The Status of Discontinuities in Brownian Risk Culture

Compared to the previous analysis which sees financial black swans as a consequence of Brownian regulation, Greenspan and Taleb's analysis sees them as unpredictable events that just fell out of the sky. Discontinuities are outliers. Discontinuities are assimilated to disordered movements of markets subject to irrational exuberance. In fact, one of the cognitive consequences of the Brownian-based risk culture is the truncation of financial time series into "normal" periods (continuous "rational" market) and periods of "insanity" where markets are deemed "irrational" (discontinuous "irrational" market). This dichotomy leaves the practitioners unable to explain the transition from one period to another. In Brownian finance, the only way to add jumps is to combine a continuous stochastic process with a stochastic jump process. Brownian finance needs the use of extreme value theory to be able to tackle the jump problem of financial risk. This is why, starting in the 1990s, extreme value theory (EVT) was rediscovered in financial risk modelling [85,86]. For some two decades, the only way to incorporate discontinuities into financial risk modelling was to use EVT. However, it is also clear how the integration of extreme value theory does not solve the problem of discontinuities at all scales of observation. Only large discontinuities are taken into account, and it is assumed that small discontinuities can be assimilated to a continuous path.

The risk culture of Brownian finance prevents itself from conceiving that small jumps are present on small scales. The assimilation of small jumps to quasi-continuity makes it impossible to measure the fragility of the market. Small jumps make it possible to evaluate the erratic structure of a stock market

path, and give an indication of its irregularity. Considering small market jumps as "micro-crashes" makes it possible to try to anticipate the occurrence of large jumps, which are, those, dangerous. Small jumps are like weak signals in the structure of financial risk. The risk culture of Brownian finance ignores small jumps because it considers them to be unimportant. In doing so, it must add extreme values to account for discontinuities.

To say it differently, the hinge of my alternative can be summarised as follows. There are two fundamentally different risk cultures for financial risk modelling. The first is "Leibnizian": It takes the continuity as a cornerstone for financial risk modelling. In this philosophical framework, following Bachelier's (1900) thesis, risk is modelled by continuous diffusion processes and jumps (market crashes, financial accidents, etc.) are added to this main process as part of another stochastic jump process. This is the "Bachelier legacy" which leads to a "diffusion + jumps" conceptual framework, a "smooth" risk culture. On the contrary, "anti-Leibnizian" position holds that discontinuity exists at all scales, even very small scales. The presence of discontinuities at all scales, even micro-scales (micro-crashes), allows grasping the profound nature of financial risk. According to this view, financial risk modelling is described by discontinuous pure jump processes, as, for instance, Lévy processes. This is the "Mandelbrot legacy", a "rough" risk culture. Mandelbrot's pivotal move of the 1960s consisted in conceptualising the discontinuity of price changes as a tenet problem for financial risk modelling.

### 4.3. A Note from Keynes

Let us conclude with a remark by Keynes, to which we will propose an answer. In *The General Theory*, Keynes stated: "The classical theorists resemble Euclidean geometers in a non-Euclidean world who, discovering that in experience straight lines apparently parallel often meet, rebuke the lines for not keeping straight—as the only remedy for the unfortunate collisions which are occurring. Yet, in truth, there is no remedy except to throw over the axiom of parallels and to work a non-Euclidean geometry. Something similar is required today in economics." I would now like to respond to Keynes's call and to do "something similar" in finance, a change of geometry for financial risk modelling leading to a new sustainable risk culture.

It is well known that Euclidian geometry is well adapted to a flat world. However, as space-time is not flat, space-time geometry is not Euclidian. A non-Euclidian geometry is geometry of curved space-time. This type of geometry has become essential for incorporating the effects of gravity when measuring radio wave frequencies: For example, the atomic clocks of the satellites in the Global Positioning System (GPS) need adjustments due to the Earth's gravitational field. In the same vein, neoclassical financial risk modelling is devised to manage risk in a one-dimensional "flat" world. In a flat world, risk has only one dimension, the volatility. Risk assessment, risk management and risk control are designed to tame and to mitigate volatility. Nevertheless, in the real world, there is another dimension to the risk, the intensity of jumps, creating discontinuities and roughness on the paths of the economic variables. For taming the jumps, risk modelling needs to "throw over the axiom" of neoclassical finance: the "flatness" of risk. To incorporate the effects of a non-flat world, it is necessary to use curved risk geometry, i.e., a new geometry including roughness at the heart of the risk models. This new risk geometry defines a new finance. Just as Euclidian geometry and non-Euclidian geometry both exist, I defined "Brownian finance" and "non-Brownian finance" which both exist, and I argue that non-Brownian finance is just as important for modelling sustainable financial risk as non-Euclidian geometry is for modelling durable travel.

Coming back to Keynes' statement ("the classical theorists [ . . . ] rebuke the lines for not keeping straight"), it is possible to show that a similar phenomenon occurred in financial risk modelling with Brownian finance. For forty years, academic research into financial risk modelling deliberately ignored the jumps, posing as an axiom a continuity assumption supposed sufficient to understand the nature of financial risk [59]. Without any validated tests, Brownian motion has become the paradigmatic stochastic process used in neoclassical financial risk modelling and built a risk culture that ignored the real risks. For this reason, the persistence of Brownian motion in financial risk modelling can be

considered as an epistemological puzzle [60]. Since the 1980s, it has been known that the performance of a portfolio is poorly explained by its average, but the average has continued to be used as a management benchmark. Even if it was established that performance was concentrated on a small number of securities or days, the average remained the unassailable benchmark for measuring risk-adjusted performance. To repeat the same sentence as previously, everything was going on as if, for mysterious reasons that had nothing to do with science, the principle of average was considered "natural" for finance and, in any case, preferable to other approaches. This is an intriguing fact which could be further explained by sociologists.

Now is the time to turn to the work of Benoît Mandelbrot.

## 5. A Possible Solution for the Sustainable Modelling of Financial Risk: Fractal Geometry

There are so many references to fractals today that it is out of the scope of this paper to give even a glimpse of them here. Only the main characteristics are presented here with the aim of introducing the possible usefulness of fractal geometry for the building of an ecological finance theory.

Let us just recall in a brief introduction that a number of "stylised facts" exist in stock price dynamics, and that these stylised facts are at odds with classical financial risk modelling [87]. To mention just a few with a minimal number of references: the distribution of return is strongly non- Gaussian with a scale-dependence kurtosis [88–91]; the return amplitudes are correlated and intermittent, creating the phenomenon known as "volatility clustering" [92–94] and it exists a non-trivial scaling on moments of the empirical distributions [95–98]. Following long and hard controversies between financial mathematics and the heterodox conceptions of financial modelling [99], the current of thought stemming from physics and known as "econophysics" contributed to a profound renewal of financial modelling based on the use of multifractal methods [100]. Multifractal analysis has been successfully validated for solving the puzzles of stylised facts since the 2000s [101–109]. The relationship between econophysics and classical mathematical finance is specified by using Sato's classification in [54].

### 5.1. Is Fractal Geometry a "Sustainable" Geometry of Nature?

In a very influential book published in 1982 that had considerable audience, *The Fractal Geometry of Nature* [110], Benoît Mandelbrot introduced the idea that one of the main characteristics of "nature" was that it possessed so-called "scaling laws" (or "power laws"), a property related to the fractal structure of nature. The word "nature" designates both physical geography and human geography. Mandelbrot's idea was that many natural patterns could be described mathematically as fractal. If the concept of "fractal" transcends the purely mathematical framework, fractal and then multifractal methods have been applied mathematically in a very large number of fields in both the physical and social sciences. It is fascinating that the same fractal structures are found in nature and in the human economy. However, they are not necessarily the same fractal "objects". I am not talking about fractal "objects" here, but about fractal structures and fractal methods.

### 5.1.1. Fractals in Nature

In fact, it seems that approximate fractal shapes are largely observed in nature. Fractals occur in an extraordinarily diverse range of phenomena. The description of "nature" (natural and man-made) by fractal geometry provides information in fields such as geology (study of relief, coasts and rivers, rock structures, avalanches), meteorology (clouds, vortexes, pack ice, rogue waves, turbulence), volcanology (prediction of volcanic eruptions, earthquakes), astronomy (structures of the universe, craters on the moon, distribution of galaxies), urban geography (urban structure, changing demographics), and economics (stock market crashes). In all these fields, fractals appear as the property of extremely irregular or fractured objects, irregularity visible at different scales. Power laws are found everywhere from molecules to trees and forests. This is the reason why applications of fractal modelling in the geosciences are huge [111–113]. Everything natural or man-made seems to be created by the laws

of fractal geometry. This geometry seems to be the signature of "how nature works" [114], like a "law of nature", a "life law" [115,116].

The word "fractal" is a neologism created by Mandelbrot in 1974 from the Latin root "*fractus*", which means broken, irregular. Mandelbrot's idea was that fractal geometry describes well the irregular face of "nature". Using fractal geometry, Mandelbrot argued, the irregular natural objects, once considered unmeasurable, could now be approached in rigorous and vigorous quantitative fashion: "Clouds are not spheres, mountains are not cones, and lightning does not travel in a straight line." Hence, "the complexity of nature's shapes differs in kind, not merely degree, from that of the shapes of ordinary geometry, the geometry of fractal shapes" [110,117,118]. A fractal pattern, or "fractal", is, in first approximation, a curve, a surface, a volume of irregular or fragmented shape that is created by following deterministic or stochastic rules involving internal homothety. A fractal is created from an initial shape that is fragmented into small pieces. Infinite repetition (iteration) is one of the essential aspects of fractal geometry. More generally, a fractal designates objects whose structure is invariant by change of scale. Fractality is another way to understand scale invariance. The infinite repetition of the same pattern is a central feature of fractals and is related to the notion of self-similarity. By zooming in or out on the object under examination, its structure is invariant. The whole looks like the part, which itself looks like a smaller part. The pattern repeats itself ad infinitum. A self-similar object is an object which is exactly or approximately similar to a part of itself (i.e., the whole has the same shape as one or more of the parts).

Some criticisms and controversies about the universality of fractal geometry addressed the issue of limited scaling ranges of data, which could invalidate the global fractal approach ad infinitum [119,120]. The answer to these criticisms was to say that the existence of a scaling range defines the validity of fractal approach for properly describing a given phenomenon [121].

### 5.1.2. Fractals and Sustainability

A possible objection to this argument is that "natural" does not always equate to "sustainable": not all "natural" things are actually "green". Many things are natural but not green, like coal, $CO_2$, etc. The difference between "sustainable" and "natural" is that sustainability emphasises the perspective of the future. According to the Environmental Protection Agency [122], "sustainability" is the study of how natural systems function, remain diverse and produce everything it needs for the ecology to remain in balance. It also acknowledges that human civilisation takes resources to sustain our modern way of life [123].

What is the relationship between sustainable development and fractal geometry? According to the Sustainable Land Development Initiative [124], people, planet and profit components of triple-bottom-line sustainability are bound by the laws of fractal geometry. This fractal geometry is revealed as follows: "Like life itself, sustainable development is built on a self-similar (but not identical) pattern that replicates itself on increasing and decreasing scales, based on a fundamental 'code'" [125,126]. If the word "green" refers to "something" related to the notion of sustainability, then, if sustainable development is based on fractal structures, then fractal geometry, as sustainable, can be considered "green". In this sense, it seems that "fractality" meets the aims and constraints of sustainability. An example is given with the scale issue in the conciliation between "local" and "global". Sustainability is associated with the conciliation of different scales and fractal geometry is describing the scaling laws [127]. A non-fractal risk culture is inherently unsustainable because it yields the illusion of a real economy without limits and without different scales.

It should also be noted that the word "sustainable" today has mainly environmental connotations, but it can also apply to financial institutions themselves. The mandates of regulators and supervisors also focus on financial sustainability: helping supervised institutions to remain solid, stable and capable of continuing to fulfil their commitments to their customers. It seems that, as time goes by, these two understandings (environmental and financial) are gradually coming together [128]. The greening of the financial system corresponds to its transformation towards generalised sustainability.

It is therefore important to be able to characterise the main features of fractal geometry in nature and economy, in order to be able to detect "how sustainable is sustainable". Appendix A presents two main properties of fractality: the 80/20 rule and the irregularity as a universal pattern of fractal structures.

*5.2. Fractalisation of Financial Risk Modelling*

Like all other models, fractals have shortcomings. The point I am arguing here is that the shortcomings of "non-fractal financial risk models" are more serious than the shortcomings of "fractal financial risk models" because of the disconnection between finance and nature.

5.2.1. The Greening of Financial Risk Models and the Green Premium Puzzles

According to several reports (e.g., European Banking Federation Report [129], "green finance" includes environmental aspects (pollution, greenhouse gas emissions, biodiversity, water or air quality issues) and climate change-related aspects (energy efficiency, renewable energies, prevention and mitigation of climate change connected to severe events). However, according to various works published on climate-related risk issues and environmental risk, the conventional financial approaches to risk are inadequate. For example, the traditional financial risk modelling is not adapted to climate risk [130] and expected climate change will require the consideration of extreme events [131] and tipping points [132]. Faced with these new situations, there is no model consensus [133]. Moreover, the results often depend on criteria other than purely technical ones, such as philosophical or ethical criteria [134–136].

Much of the current work seeks to develop methodologies for analysing financial risk models that incorporate environmental and social concerns (the "greening" of the financial risk models). Currently, an increasing number of investors are adopting "impact covenants", i.e., legally binding commitments, as tools to maximise the positive impact of their investments, with the use of neoclassical risk modelling approaches oriented with green stakes. However, recent work argues that teaching the classical metrics for new stakes is unethical [137,138]. In these debates, my proposal is to consider fractal geometry for the construction of green metrics. I suggest that new green metrics could avoid the split between the neoclassical structure of risk modelling and impact covenants.

At this point, it is important to emphasise something non-trivial in order to fully understand how ecological finance theory is different from green finance. The neoclassic way to evaluate a green project is to change the parameters of the evaluation by adding green attributes. This is a financial strategy that nudges the market by "greening" neoclassical metrics. This way of doing things has been noticed in [5] (n. 14), where it is said that recent research on "sustainable finance" superimposes a neoclassical financial risk modelling to the analysis of sustainability concerns. This kind of sustainable finance analyses new ecological concerns through the lens of technical tools based on Brownian finance, rather than questioning the foundations of neoclassical risk modelling to respond to the sustainability challenge. It is important here to clearly understand that the fractal way of risk modelling is not a neoclassical risk modelling "tinkering" with "green parameters", but a new epistemological framework to design new tools and new regulations built for new ecological stakes. The "green bonds premium puzzle" [139] can be seen as a trace of this confusion between neoclassical greened metric and fractal metric.

For example, in the case of the financial valuation of a "green project", in order to take into account the fractal nature of the project's risks for its funding, the fractal characteristics of the environment will have to be measured, and then a new measure, a "green metric", will have to be defined for the valuation of the project. In this case, it will not be a question of modifying the discount rate or the numbers of the future values and future risks of the green project, but of modifying the calculation technique itself, independently of the parameters chosen to feed the valuation models. A green metric is not a neoclassical metric greened by green parameters, but a metric created with entirely new fundamental principles based on the fractal characteristics of nature.

5.2.2. How to Reconcile Finance and Nature

Recalling again that, in finance, mathematical models are not a "part of reality" but the generator of that reality [11,31–33,36], the issue for designing sustainable financial risk modelling could be: With which financial generator do we want to perform the real world? We have seen above that, according to some work, this performative power of flawed financial risk modelling can be understood as the "financial ontology" of "real" catastrophes [140].

In this section, I argue that considering fractal geometry—even with its shortcomings—could mitigate the financial ontology of catastrophes because fractal models of financial risk are a better fit with both the "nature" of the real world and the "nature" of the man-made world of economy. From the perspective of the connection between financial risk and environmental risk, fractal modelling of financial risk can help to reconcile nature and finance. In particular, non-fractal financial risk models cannot be aligned with fractal natural risk models. This disconnection has been one of the causes of the financial *hubris* and recurring financial failures since 1987. If environmental risk can be properly modelled with fractals, and financial risk is modelled without fractals, then finance can "slip" over nature and, if necessary, damage nature by creating the illusion that no environmental risk exists.

It is precisely because of the misalignment between the fractal models of environmental risk and the non-fractal descriptions of financial risk that the disconnection of finance from the economy was made possible. This disconnection came about because we wanted to cover the rough with the smooth. At the risk of repeating even more, the financial *hubris* stems from the desire to hide the roughness with smoothness. With the roughness modelling of fractal financial risk models, the financial ontology of natural catastrophes would be replaced by the natural ontology of financial risk. According to Mandelbrot [141], "The reasons are that the main feature of price records is roughness and that the proper language of the theory of roughness in nature and culture is fractal geometry." The core of Mandelbrot's argument is the following: If the relevant description of nature is fractal, then relevant financial risk modelling will have to be anchored on fractal geometry.

If price changes have fractal properties [142–146], then it is a matter of matching these fractal properties to the fractal properties of nature: Financial fluctuations will be aligned with changes in the natural quantities of the economic, geographic or human environment. In this sense, the fractalisation of financial risk modelling can become the new frontier for ecological finance theory: "even when the present fractal models become superseded, fractal tools are bound to remain central to finance" [141]. Using fractals to model financial risk is a necessary but not sufficient condition to build sustainable financial risk modelling.

## 6. Conclusions and Future Research

In this conceptual contribution, I challenge the common diagnosis of the 2008 financial crisis and the reasons why neoclassical finance theory is unsustainable: greed and the absence of ethics and environmental values. I point out the weaknesses of this approach and the unfulfilled promises of the "greening" of financial models, and I argue for research to move forward with a different philosophy. I propose to go back to the background philosophical framework of neoclassical finance theory and identify both the Leibniz scheme and the lessons that financial modelling should draw in order to minimise the possibility of a prejudicial and unsustainable dominance of risk models in the future. I then present a possible alternative approach to financial risk modelling that is aligned with environmental risk to improve the sustainability of finance from a global perspective.

The possible alternative is the following: I hypothesise that if the relevant description of "nature" (human and physical geography) is fractal, and if to be in line with nature allows to become sustainable, then sustainable financial risk modelling has to take account of a specific "signature" of nature—its fractal geometry. Consequently, sustainable financial risk modelling has to include fractal properties in the mathematical structure of risk models. Scaling laws seem appropriate tools adapted to the description nature (it explains the "law of 80/20"), and the fundamental brick to reconstruct a financial theory tailored to natural resource constraints.

Assessing the morphology of randomness underlying the technical approaches of neoclassical finance allows me to consider that the unsustainability of neoclassical finance theory was due to the lack of consideration of fractality in financial risk modelling, creating a faulty risk culture. In this line, the main problem of neoclassical financial risk modelling is due to the widespread use of the continuity principle and the theory of averages, i.e., two non-fractal representations of nature, a paradigm of a "smooth" world. I have defined two risk cultures as two types of financial risk geometries: Brownian finance and non-Brownian finance. Finally, starting from a simplified presentation of the fractals, I tried to lay the ground for the fractalisation of financial risk modelling with the objective of contributing to the future development of Ecological Finance Theory by providing a first insight into the usefulness of fractals for SDGs.

I took the opportunity of the special issue of *Sustainability* on "Finance and Agenda 2030" to push financial theory forward by first criticising the mainstream approaches and then presenting a different alternative approach, which does what the mainstream approaches (greening of financial instruments) do not do, while maintaining the common underlying logic of risk fractality present in international regulations, Basel III and Solvency II. There are, however, many other alternatives that are worth exploring. The objective of this paper was only to draw attention to the importance of augmenting mainstream thinking (the greening of neoclassical financial theory with the greening of investor preferences) to explore what appears to be a promising avenue of multifractal methods to design new tools for sustainable financial risk modelling.

Let us just reiterate that the intention here is not to add further hard proof to validate fractals and multifractals, but to propose a new research programme, a "new agenda" for future research. Fractals and multifractal methods are identified as a possible pathway for sustainable risk modelling, but the aim is not to deepen fractal and multifractal modelling or to test fractal and multifractal modelling on real data, which has already extensively been done elsewhere, but merely to propose a possible avenue for research in a heuristic manner. I simply wish to set out an agenda for future research by discussing what the main steps of this pathway might be. My proposition is as follows:

(1) To verify how good are fractal and multifractal methods to model important features of physical and human geographies in the sense of SDGs.
(2) If a relevant financial risk modelling should take these fractal characteristics into account, to seek to construct "green metrics" and tools in the fractal sense to lay the groundwork for ecological finance theory as an alternative of neoclassical finance.
(3) To apply these new metrics to portfolio management and risk measurement techniques.
(4) To introduce financial risk modelling issues and metrics into a philosophical reflection on epistemic ethics because financial tools and financial risk modelling contribute to shape the real world. This last issue is important, keeping in mind that ecological finance theory aims to move from "what is" to "what should be".

**Funding:** This research received no external funding.

**Acknowledgments:** I would like to thank the two reviewers for their very helpful comments and questions, which helped to significantly improve this article. Special thanks to Thomas Lagoarde-Segot for encouraging me to prepare this work and for sharing the project on Ecological Finance Theory. My warms thanks also go to Jean-Philippe Bouchaud, Eve Chiapello, Olivier Le Courtois and Emmanuel Picavet for always providing very fruitful and stimulating conversations. This work also benefits from the numerous conversations I have had on the necessity of rebuilding finance on a new foundation, with Christophe Revelli and Christophe Faugère at Kedge Business School, and the CSR team at the University Paris 1 Panthéon-Sorbonne (ISJPS, UMR 8103), in particular, Kathia Martin-Chenut. Parts of this work were presented separately in the AFIR, ARC, QMF, INFINITI, SASE and HES conferences between 2014 and 2019. I would like to thank the participants at these conferences for the questions and discussions. The Fondation Maison des Sciences de l'Homme has, as always, been a very fertile workplace for the quest for new ideas and provided exactly the right ambience and cultural mix of colleagues needed for developing most of the ideas that are presented here.

**Conflicts of Interest:** The author declares no conflict of interest.

## Appendix A

*Pareto Distributions, Scaling Laws and Irregularity*

A convenient way to get into fractal geometry is to start with what everyone knows and experiences every day: The adage that "very few have much and many have very little". This concentration of a phenomenon on its extremes can be found everywhere in the geosciences and in the real economy. The Pareto distribution of income (the so-called "law of 80/20") is a canonical example of this. The "law of 80/20" states that for many phenomena, about 80% of the total (e.g., total portfolio gain, liquid mass of water on planet Earth) comes from about 20% of the population (e.g., very few stocks in the portfolio, very few oceans or large lakes on planet Earth). There are so many natural or economic phenomena that follow this law that it is impossible to give even an initial reference list here. What all these phenomena have in common is that very few oceans or large lakes concentrate most of the water on Earth, very few titles concentrate most of the total gain, but also very few scientific articles concentrate most of the quotations, very few words are found very often in speeches (the so-called Zipf's law), very few books are sold in very large numbers, and so forth [147–149]. The Lorenz curve represents this phenomenon on a graph whose coordinates are the percentage of the population concerned and the percentage of the total obtained.

Power laws have the property to be "scale invariant". That means that scaling the quantity by a constant c multiplies the power law relation by $c^{-\alpha}$. If the scale invariance property is validated, objects do not change if scales of length are multiplied by a common factor. It represents a kind of universality. Like Russian dolls, the shape of the object repeats itself at all sizes. It is said that there is a self-similarity of the phenomenon being studied. So there is a relationship between power law and self-similarity. Here again, in practice, there are several forms of self-similarity, but we do not want to complicate the presentation in the context of this introductory article.

Let us express this phenomenon of concentration by describing it with the cumulative distribution function of the population under consideration. To understand it intuitively, one has to ask oneself the question: How many values are greater than or equal to a given value? Using probabilities, the question is $\Pr(X > x)$ for $x > a$ where $a$ is a given threshold. $\Pr(X > x) = 1 - F(x)$ is the opposite of the cumulative distribution function (also called survival function or tail function). In the case of a phenomenon with a high concentration of the total on the largest values of the distribution beyond a given threshold, we find that $\Pr(X > x) = (a/x)^\alpha$ for $x > a$. We find a power law. The name of this function comes from the fact that one of the quantities (here, the cumulative empirical frequency) varies as the power of another (here, the values exceeding the threshold). This is the Pareto type I distribution. A very large number of situations are possible, especially with other power laws, but we do not intend to make the presentation more complicated here. Turning now to the empirical data and empirical frequencies, a rank-frequency diagram allows us to visually capture a characteristic property of Pareto's power laws. The straight line of the log-log plot has the slope $-\alpha$. In fact, when logarithms are taken instead of raw values, if $Y = a\,X^{-\alpha}$, then $\ln Y = -\alpha \ln X + k$. We deduce from this that the "signature" of a power law is a linear relationship between the two quantities in log-log diagram.

The parameter $\alpha$ is the exponent of Pareto's distribution, the slope of the log-log line, and indicates the degree of concentration of the distribution. The value of the parameter $\alpha$ determines the existence of the moments of the distribution. The variance (central moment of order 2) exists if and only if $\alpha > 2$ and the mathematical expectation exists if and only if $\alpha > 1$. If $\alpha < 2$, the variance does not exist (it is infinite). If $\alpha < 1$, the mathematical expectation does not exist (it is infinite). It can be seen from this elementary example that, in the case of high concentration distributions, the variance, or mathematical expectation, may not exist. Averages can always be calculated, but since the theoretical values of moments are infinite, the averages will be unstable or diverge. Moreover, they will be sensitive to the larger values of the distributions. Thus, Pareto's distributions invalidate the simple use of Quetelet-style averages in risk modelling.

Let us now move from a Paretian representation of unequal distributions of static phenomena (oceans, book sales, etc.) to the analysis of a dynamic phenomenon with the same concerns. Let us consider the values, no longer of record sales or the liquid volume of lakes, but of variations in a quantity that we follow over time (rainfall, river level, wind speed etc.). We will find on these variations the same characteristics as on the static samples: concentration of the total variation on the strongest variations. If we then consider the shape of the trajectory examined, we can guess that the more the variations are concentrated on a few very large values, the more irregular the trajectory appears. It can be seen that the irregularity is closely related to the concentration property, and that the greater the concentration, the greater the irregularity will be, i.e., (in the previous example) the parameter $\alpha$ is small. In the absence of variance, and even a mean, the irregularity will be extremely large. At certain moments, the path will give the impression of "jumping" strongly, and large discontinuities will appear. There is, therefore, a relationship between concentration, scaling laws, discontinuity and irregularity.

## Appendix B

*Fractal Properties of Neoclassical Finance Theory and Beyond*

How to fractalise the financial risk modelling? Anew, the question is so vast that it would be impossible here to even summarise the work that has been undertaken on this subject. To paint a general picture, let us say that, in order to model financial risk as fractal, we must quantify the price changes with fractal properties. The simplest way to consider a price change is to use the distribution of changes $\Delta X$ over a given time scale $\Delta t$. If price fluctuations can be described by fractal properties, then we need to search for fractal properties on the distributions of price changes $\Delta X$ at different scales $\Delta t$. The statistical structure of price change $\Delta X$ has to be described at different scales (small and large $\Delta t$). To be accurate, let us say that $\Delta X(t)$ is the log-difference of prices $S$, i.e., $\ln S(t)$—$\ln S(t-1)$ for normalised time intervals. To say it differently, $X(t)$ is the continuous log-return of the stock price $S$ between time 0 and time $t$, i.e., $X(t) = \ln S(t)$—$\ln S(0)$. In the standard model of stock market variations (neoclassical finance theory), $X(t)$ follows a Brownian motion with trend $\mu$ and standard deviation $\sigma$, which is the "volatility" of price fluctuations. The standard model of stock market variations is $X(t) = \mu t + \sigma W(t)$ where $W(t)$ is a Wiener process, i.e., $W(0) = 0$ and $\mathbf{E}(W_1) = 1$. In the standard model of stock market variations, the "square-root-of-time" rule describes a scaling law on volatility. This scaling law allows annualising the volatility from monthly measures of empirical volatility.

More generally, the issue arises of moving from one time horizon $t$ to another time horizon $at$, i.e., a time horizon which is a multiple of $t$. The mathematical formulation of this problem is as follows. Let $\lambda(t)$ be a parameter of the distribution of $X(t)$. The objective is to determine the parameter $\lambda(at)$ of the distribution of $X(at)$. Among the parameters useful to qualify the risk of a distribution are the moments of order $k$, the $k$-th moments. The moments give an idea of the shape of the distribution (flat, peaked, symmetric, asymmetric, etc.). In particular, if $m_k(X, t)$ is a central moment of order $k$ of the random variable $X(t)$, the presence of a scaling law on $X(t)$ implies the existence of a scaling law on moments. Hence, the issue is to obtain the central moments $m_k(X, at)$ at a scale $at$ knowing the central moments $m_k(X, t)$ at scale $t$. This is given by using the convolution product of the distributions of random variables $\Delta X$. Therefore, we see that the problem of the scaling laws of financial risk is closely linked to the issue of scale invariance of risk: Is the shape of the distribution the same on different time scales? If the answer is "yes", hence, there is a fractal property of financial risk.

In Brownian finance, as the distribution used is Gaussian, there is a fractal property of volatility: the shape of financial risk remains Gaussian at every scale. This is the fractal property of Brownian motion. It means that, implicitly but clearly, in neoclassical finance, risk is assumed fractal because the volatility is supposed to follow the celebrated "square-root-of-time-rule" [150]. This represents the time-scaling of risk in neoclassical finance. The "square-root-of-time-rule" in finance states that the annual volatility is given by the monthly volatility multiplied by the square root of the duration measured in months, i.e., 12. More generally, in neoclassical finance, the risk is scaled by the square root

of the time horizon. This relation describes a scale invariance property of the Brownian motion used in neoclassical risk modelling. The annual risk has the same shape as the monthly risk: a Gaussian distribution. It can therefore be seen that the fractality of risk is not in itself sufficient to align finance with nature, since the equations of neoclassical finance contain a fractal property. Some fractals are relevant, some are not. The fractalisation of risk modelling has to deal with the determination of relevant fractals.

The presence of discontinuities at every scale in financial time series gives a light on this puzzle. Fractals built with a Gaussian base are not rough enough. It is necessary to capture the roughness of stock market paths to avoid financial *hubris*. Examples of rough fractals with discontinuities can be found in nature. It is then a matter of transposing the fractal characteristics of nature to the fractal characteristics of financial risk. A risk fractality adjustment can be made by measuring the fractality of the real phenomenon being investigated in the economy.

To give a first and elementary idea of what the fractalisation of a risk can be, let us consider the relation of neoclassical finance on the central moment of order 2, the variance. The variance is—according to the definition of the Brownian motion—linear in time, therefore, we have:

$$m_2(X, at) = a \times m_2(X, t) \tag{A1}$$

From which comes the "square-root-of-time-rule" on volatility which is:

$$\sigma(X, at) = \sqrt{a} \times \sigma(X, t) \tag{A2}$$

This is the reason why:

$$\text{volatility}(X, 1\,\text{year}) = \sqrt{12} \times \text{volatility}(X, 1\,\text{month}) \tag{A3}$$

Considering the quantity called "Value-at-Risk" (VaR), we will have in the same way:

$$\text{VaR}(X, 10\,\text{days}) = \sqrt{10} \times \text{VaR}(X, 1\,\text{day}) \tag{A4}$$

These are exactly the capital requirement calculations recommended by the Basel III and Solvency II regulations. We see how neoclassical finance unconsciously uses fractals—but not rough fractals—smooth fractals.

If now the fractal characteristic of financial risk is not the one defined by the fractal properties of Brownian motion, we can consider that there will be another scaling law on volatility, which will be:

$$\sigma(X, at) = a^H \times \sigma(X, t) \tag{A5}$$

$H$ represents the fractality exponent. $H$ will be obtained by the "natural" properties of the object (for example, properties given in geophysical records) whose financial risk is to be modelled. The "square-root-of-time-rule" would be, in this case, a "time-power-$H$-rule". In the simplest case, $H$ is a constant. However, we can alternatively have $H(t)$ which would mean that the exponent of fractality varies with time:

$$\sigma(X, at) = a^{H(t)} \times \sigma(X, t) \tag{A6}$$

The scaling property is not the same as previously. As we can see in this elementary overview, introducing fractality in financial risk modelling is like transforming some important constants of neoclassical finance into variables that introduce roughness where previously there was smooth.

In most cases, and in particular in the basic example given above, the simple fractals corresponding to the first Mandelbrot models are not sufficient to account for the complexity of the phenomenon to be modelled. Let us give a brief overview of Mandelbrot's models. Mandelbrot developed his fractal models in two strands of papers. A good introduction to the usefulness of his models for finance is

given in *Fractals and Scaling in Finance* [151]. Let us just quickly say that Mandelbrot's new way of modelling price changes was based on concentration, discontinuities, scaling laws and time-change processes. Mandelbrot designed several models of fractals for finance, and presented them in several overlapping layers of articles or chapters, often intertwined with each other, with back and forth between models through time. So the financial fractal landscape is quite complicated to explore (in a sense, it is itself fractal).

To put it in a nutshell, there were two distinct problems to deal with: that of heavy distribution tails (high probability of large risks) and that of long-range dependence (long memory of stock market movements) [152]. Mandelbrot's early models dealt with the two problems separately. The 1962 model resolves the issue of heavy distribution tails but within a framework of independent market movements, so the question of long dependence remains. The 1967 model deals with the question of long dependence but in a Brownian framework, so the question of heavy non-Gaussian tails remains. Finally, in a third model in 1972, Mandelbrot combined the two issues, that of distribution tails and that of long dependence, by assuming that market time did not follow the time of the calendar clock. Market time was like an intrinsic time, a time different from physical time. Using multifractal time, Mandelbrot showed that it was possible to model financial risk with a Brownian motion transformed by a time-change. A heuristic mathematical presentation of the time-change is the following. Instead of $X(t) = B(t)$ where $B(t)$ is a standard Brownian motion, we have $X(t) = B(T(t))$ where $T(t)$ is the trading time. This third model represents the transition from unifractal models to multifractal models. Finally, these ideas were synthesised in a series of three articles published in 2001, which represented Mandelbrot's comeback to the financial risk modelling arena and the new statement of "Mandelbrot's programme" [54].

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
