# Peer review of "Sustainable Financial Risk Modelling Fitting the SDGs: Some Reflections"

_sustainability, doi:10.3390/su12187789_

Round 1

Reviewer 1 Report

I’ve mixed feelings concerning the reviewed article. Generally, it’s well written, there are many references and even some ideas which could stimulate discussion in the paper. It can be interesting for socioeconomists, politicians or managers. However, from the point of view of mathematicians (or financial mathematics), it is rather useless. The author concentrates on the critique of “Brownian finance” or “Brown finance” (as it is called by the author) and proposes other approach based on fractals. However, the shortcomings of application of the Brownian motion in option pricing (and other areas of financial mathematics) are widely known and there are many articles which try to overcome these problems. The author proposes another model, but:

  1. We know that each model is only a part of reality, and each model has its shortcomings. What about the possible shortcomings of fractals?
  2. In the reviewed paper, there are only some propositions (the word “if” is used many times), the proposed model is not proved, analysed and compared with the “classical approach”. There is no evidence in the article, that this model is better or even “closer” to the real-life data. We can talk about ideas, but in the case of financial mathematics, some “hard proofs” are necessary. Fractals are an important part of nature, but not the only law of nature (e.g., can we use gravity, which is “very natural”, as the part of financial modelling?).
  3. Moreover, the author tries to connect “fractalization” of financial risks with “green finance” and “ecological finance theory”. I’m sorry, but this connection is a very week for me. Even if “fractalization” is a better way to model financial risks (compared with the more classical approach), why it can be called “green” in any way? Because it is better? The author tries to explain this connection, e.g. he states that fractals as a part of nature are “natural” (therefore, they are “green” because “to imitate the nature can save the nature”). We know many things, which are “natural but not green” (i.e. coal, CO2 etc.). Using fractals to model geophysical records is rather far from modelling “the financial risks in harmony with these characteristics” and its applications in “green economy”.

Reviewer 2 Report

The manuscript argues that there is an urgent need to rebuild finance on an ecological basis, to re-embed financial systems within ecological constraints and to develop an “ecological finance theory”. Moreover, the manuscript states that if the relevant description of nature is fractal, then financial risk modelling will have to be anchored on fractal representations in order to be in line with nature and become sustainable. More specifically, the author claims that finance needs to be "detoxified" from the use of continuous stochastic processes and of mathematical expectation.

I find the author's view that finance need to be rebuilt to be much too excessive. Obviously, current theory can and will be improved over time, but there is no reason to discredit the entire field. Naturally, financial models build on numerous assumptions, which we all know are not realistic. For example, we often assume no taxes, similar interest rate for lenders and borrowers, no constraints on short sales, etc. So, should we through away all models, just because these assumptions do not hold in reality? Although we know that this may make them inaccurate, still, these assumptions enable us to construct solvable models, which provide valuable insights. Therefore, I believe that the argument that in reality processes are discontinuous therefore we should throw away most of the financial work is unfounded and unconstructive.

Another comment is that current research in finance, specifically dealing with financial risk, is well aware of discontinuity. Several published papers present models that consider it. Therefore, this point is not new. See, for example:

  • Boudt, K., Croux, C., & Laurent, S. (2011). Robust estimation of intraweek periodicity in volatility and jump detection. Journal of Empirical Finance18(2), 353-367.
  • Liu, G., & Hong, L. J. (2011). Kernel estimation of the Greeks for options with discontinuous payoffs. Operations Research59(1), 96-108.
  • Wang, X., & Tan, K. S. (2013). Pricing and hedging with discontinuous functions: Quasi–Monte Carlo methods and dimension reduction. Management Science59(2), 376-389

Round 2

Reviewer 1 Report

The author has improved the paper in some areas, however, my general conclusion (from the previous review) is still valid. If the article is intended for philosophers, politicians, managers etc., then it could be interesting for them. In this case, it can be published (“minor revision”) when the previously mentioned problem of the connection between “fractalization” of financial risks and “green finance” is explained in a better way. The author has tried to do this, but this problem still lacks proper justification in my opinion. Some additional votes for (apart from the sentences authored by Mandelbrot) should be found and cited. If the paper is intended for mathematicians (or experts in financial mathematics) then without deeper mathematical/statistical analysis of the considered foundations (like comparing the results for “Brownian finance” with these for “non-Brownian finance”, their shortcomings, possible predictions etc.), the paper is useless (“reject”) because some general ideas are only discussed without necessary “hard proofs”.

Reviewer 2 Report

I appreciate the efforts the author made to answer my comments and to revise to manuscript. However, I still have significant concerns regarding the contribution of the paper and its scientific validity. Mainly, it is not clear whether the main purpose of the manuscript is to argue that financial risk modelling lays on problematic assumptions, or to encourage the use of fractals in finance. If it is the former, than, I as wrote in my previous report, it is not a new criticism. Previous papers addressed this issue, both the ones noted by the author, and the ones I mentioned in my prior report. Even though the author notes in his response that the manuscript deals with "… drawing more attention to the epistemological background of mathematical risk modelling than to the mathematical modelling itself", the distinction between the two is not clear. Again, the limitations of the "paradigm" are well-known.

However, if the main purpose of the manuscript is to argue in favor of using fractals, than this argument lacks sufficient foundation. Economic and financial concepts are man-made. Therefore, it is not clear why the author assume that they should follow patterns found in nature, such as fractals. Hence, I do not find any justification to the following statement (p. 2), which is central to the manuscript: "financial risk modelling will be sustainable if it fits to the basic principles of biomimicry with fractals: nature as model, nature as measure, and nature as mentor (Benyus, 1997). To put it in a nutshell, if the relevant geometry of nature is fractal, then financial risk modelling will have to be anchored on fractal representations in order to be in line with nature and become sustainable."

Round 3

Reviewer 1 Report

The author has improved the paper in the areas, which were mentioned in my previous review.

Author Response

Thank you for your report. Final revisions have again been made to respond to reviewer 2. There is a new title to better reflect the content of the paper.

Reviewer 2 Report

Regretfully, I believe the manuscript still lacks contribution. As I noted in my previous review report, the criticism on the continuity assumption is not new. The author argues in his response letter that "Old-fashioned critics only address the statistical or mathematical framework of financial risk models  …. they cannot explain the persistence of Brownian representation in financial risk modelling during 50 years while all statistical tests invalidated it." Well, the manuscript does not explain it either. The current version mentions terms such as mental model and philosophy of chance, but does not address their constructs and how they apply to this case. Therefore, the manuscript does not explain why financial academics still use the Brownian motion assumption. Moreover, taking into account that better financial models translate to higher profits, does it really make sense that a mental model prevents financial experts from using more accurate models, relieved from the Brownian motion assumption?

Notably, the third version still completely ignores financial papers offering alternative models (some of which I cited in my first review report).

Another major issue is that the linkage between better financial models and "green finance" is still not clear – does the author believe that any assumption that does not hold in reality (such as no taxes, equal interest for borrowers and lenders, no short constraints) makes the model "unsustainable"? Does the author hold that models relived of the Brownian motion assumption, but based on other unrealistic assumptions, are "sustainable"?

Author Response

This manuscript is a resubmission of an earlier submission. The following is a list of the peer review reports and author responses from that submission.